# Towards understanding retrosynthesis by energy-based models

Ruoxi Sun[1], Hanjun Dai[2], Li Li[3], Steven Kearnes[3], and Bo Dai[2]

[1]Google Cloud AI [2]Google Brain [3]Google Research
{ruoxis, hadai, leeley, kearnes, bodai}@google.com

## Abstract

Retrosynthesis is the process of identifying a set of reactants to synthesize a target molecule. It is critical to material design and drug discovery. Existing machine learning approaches based on language models and graph neural networks have achieved encouraging results. However, the inner connections of these models are rarely discussed, and rigorous evaluations of these models are largely in need. In this paper, we propose a framework that unifies sequence- and graph-based methods as energy-based models (EBMs) with different energy functions. This unified view establishes connections and reveals the differences between models, thereby enhances our understanding of model design. We also provide a comprehensive assessment of performance to the community. Additionally, we present a novel dual variant within the framework that performs consistent training to induce the agreement between forward- and backward-prediction. This model improves the state-of-the-art of template-free methods with or without reaction types.

Retrosynthesis is a critical problem in organic chemistry and drug discovery [1–5]. As the reverse process of chemical synthesis [6, 7], retrosynthesis aims to find the set of reactants that can synthesize the provided target via chemical reactions (Fig 1). Since the search space of theoretically feasible reactant candidates is enormous, models should be designed carefully to have the expression power to learn complex chemical rules and maintain computational efficiency.

Recent machine learning applications on retrosynthesis, including sequence- and graph-based models, have made significant progress [3, 8, 9]. Sequence-based models treat molecules as one-dimensional token sequences (SMILES [10], bottom of Fig 1) and formulate retrosynthesis as a sequence-to-sequence problem, where recent advances in neural machine translation [11, 12] can be applied. In this principle, the LSTM-based encoder-decoder frameworks and, more recently, transformer-based approaches have achieved promising

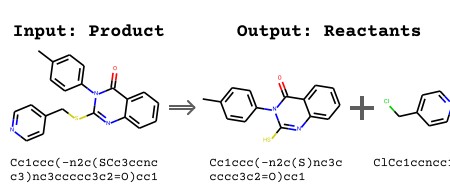

Figure 1: Retrosynthesis and SMILES.

results [13, 12, 14]. On the other hand, graph-based models have a natural representation of human-interpretable molecular graphs, where chemical rules are easily applied. Graph-based approaches that perform graph matching with templates (e.g. chemical rules) or reaction centers have reached encouraging results. Among those, G2Gs [15], RetroXpert [16] and GraphRETRO [17] outperform template-based methods by inferring reaction centers in a supervised way. In this paper, we focus on one-step retrosynthesis, which is also the foundation of multi-step retrosynthesis [3].

Our goal here is to provide a unified view of both sequence- and graph-based retrosynthesis models using an energy-based model (EBM) framework. It is beneficial because: First, the model design with EBM is very flexible. Within this framework, both types of models can be formulated as different

35th Conference on Neural Information Processing Systems (NeurIPS 2021).

EBM variants by instantiating the energy function into specific forms. Second, EBM provides principled ways for training models, including maximum likelihood estimator, pseudo-likelihood, etc. Third, a unified view is critical to provide insights into different EBM variants, as it is easy to extract commonalities and differences between EBM variants, understand strengths and limitations in model design, compare the complexity of learning or inference, and inspire novel EBM variants. To summarize our contributions:

- We propose a unified energy-based model (EBM) framework that integrates sequence- and graph-based models for retrosynthesis. To our best knowledge, this is the first effort to unify and exploit inner connectivity between different models.

- We perform rigorous evaluations by running tens of experiments on different model designs. Revealing the performance to the community contributes to the development of retrosynthesis models.

- Inspired by such a unified framework, we propose a novel generalized dual EBM variant that performs consistent training over forward and backward prediction directions. This model improves the state-of-the-art by $4.3\%$.

## 1 Energy-based model for Retrosynthesis

Retrosynthesis is to predict a set of reactant molecules from a product molecule. We denote the product as $y$, and the set of reactants predicted for one-step retrosynthesis as $X$. The key for retrosynthesis is to model the conditional probability $p(X|y)$. EBM provides a common theoretical framework that can unify many retrosynthesis models, including but not limited to existing models.

An EBM defines the distribution using an energy function [18, 19] . Without loss of generality, we define the joint distribution of product and reactants as follows:

$$p_\theta(X, y) = \frac{\exp(-E_\theta(X, y))}{Z(\theta)} \qquad (1)$$

where the partition function $Z(\theta) = \sum_y \sum_X \exp(-E_\theta(X, y))$ is a normalization constant to ensure a valid probability distribution. Since the design of $E_\theta$ is free of

---

**Algorithm 1** EBM framework

**[Train Phase]: Learning**
**Input**: Reactants $X$ and products $y$.
**1.** Parameterize $X$ and $y$ in *Sequence* or *Graph* format.
**2.** Design $E_\theta$ {e.g. dual, perturbed, bidirectional, graph-based, etc} // Sec 2
**3.** Select training loss to learn $E_\theta$ and obtain $\theta^*$ // Sec 3
**Return** $\theta^*$
**[Test Phase]: Inference** // Sec 4
**Input**: $\theta^*$, $y^{\text{test}}$, Proposal $P$. // Sec 4
**4.** Obtain a list of $X$ candidates by $P$.
$L^{test} \leftarrow P(y^{test})$
**5.** $X^* = \arg\min_{X \in L^{\text{test}}} E_{\theta^*}(X, y^{\text{test}})$
**Return**: $X^*$

---

choice, EBMs can be used to unify many retrosynthesis models by instantiating the energy function $E(\theta)$ with various designs. Note there is a trade-off between model expression capacity and learning tractability. EBM is also easy to obtain conditioning with different partition functions. The forward prediction probability for reaction outcome prediction $p_\theta(y|X)$ can be written as $\frac{\exp(-E_\theta(X,y))}{\sum_{y'} \exp(-E_\theta(X,y'))}$ with the same form of energy function.

The proposed framework works as follows: **Step 1**, design and train an energy function $E_\theta$ (Sec 2 and Sec 3), and **Step 2** use $E_\theta$ for inference in retrosynthesis (Sec 4). See Fig 2 and Algorithm 1.

## 2 Model Design

Based on how to parameterize reactant and product molecule $X$ and $y$, the model designs can be divided into two categories: sequence-based and graph-based models.

### 2.1 Sequence-based Models

Here we describe several sequence-based parametriztion to instantiate our EBM framework, which use SMILES string as representations of molecules. We first define the sequence-based notations.

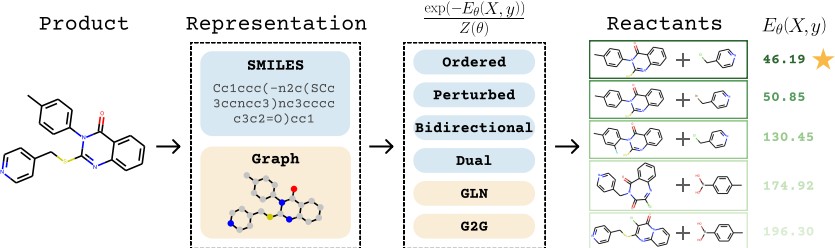

Figure 2: **EBM framework for retrosynthesis**. Given the product as input, the EBM framework (1) represents the product molecule as SMILES sequence or a graph, (2) designs and trains the energy function $E_\theta$, (3) ranks reactant candidates with the trained energy score $E_{\theta^*}$, and (4) identifies the top $K$ reactant candidates. The best candidate has the lowest energy score (denoted by a star). The list of reactant candidates is obtained via templates (template based proposal) or directly generated by the trained model (template free proposal).

Given a reactant molecule $x$, we denote its SMILES representation as $s(x)$. Superscript $s(x)^{(i)}$ denotes the character at $i$-th position of the SMILES string. For simplicity, we use $x^{(i)}$ when possible. Reactants of a chemical reaction are usually a collection of molecules: $X = \{x_1, x_2, .., x_j, .., x_{|X|}\}$, where $x_j$ is the $j$-th reactant molecule. The SMILES representation of a molecule set $X$, denoted as $s(X)$, is a concatenation of $s(x)$ for every $x$ in $X$ with "." in between: "$s(x_1).s(x_2)...s(x_{|X|})$". We use $X^{(i)}$ as the short form of $s(X)^{(i)}$ to denote the $i$-th position of the concatenated SMILES.

### 2.1.1 Full energy-based model

We start by proposing a most flexible EBM that imposes the minimum restrictions on the design of $E_\theta$. All the variants proposed in Sec 2.1 are special instantiations of this model (e.g. by specifying different $E_\theta$). The EBM is defined as follows:

$$p(X|y) = \frac{\exp\left(-E_\theta(X, y)\right)}{\sum_{X' \in \mathscr{P}(M)} \exp\left(-E_\theta(X', y)\right)} \tag{2}$$

$$\propto \exp(-E_\theta(X, y)) \tag{3}$$

Here the energy function $E_\theta : \mathscr{P}(M) \times M \mapsto \mathbb{R}$ takes a molecule set and a molecule as input, and outputs a scalar value. $M$ defines the set of all possible molecules. $\mathscr{P}(\cdot)$ represents the power set. $\mathscr{P}(M)$ denotes domain of reactant sets $X$. Due to the intractability of the partition function, training involves additional information e.g., template or approximation of the partition (See Sec 3).

### 2.1.2 Ordered model

One design of energy function is factoring the input sequence in an autoregressive manner [12, 20].

$$p_\theta(X|y) = \exp\left(\sum_{i=1}^{|s(X)|} \log p_\theta(X^{(i)}|X^{(1:i-1)}, y)\right) \tag{4}$$

$$= \exp\left(\sum_{i=1}^{|s(X)|} \log \frac{\exp\left(h_\theta(X^{(1:i-1)}, y)^\top e(X^{(i)})\right)}{\sum_{c \in S} \exp\left(h_\theta(X^{(1:i-1)}, y)^\top e(c)\right)}\right) \tag{5}$$

where $p_\theta(X^{(i)}|X^{(1:i-1)}, y)$ is parameterized by a transformer $h_\theta(p, q) : S^{|p|} \times S^{|q|} \mapsto \mathbb{R}^{|S|}$ where $S$ is vocabulary. $e(c)$ is a one-hot vector with dimension $c$ set to 1. This choice of $h_\theta(p, q)$ enables efficient computing of the partition function, as it outputs a vector with length equal to $|S|$ to represent logits (unnormalized log probability) for each value in vocabulary. Here, maximum likelihood estimator (MLE) is feasible for training, as this factorization allows tractable partition function.

### 2.1.3 Dual model

A different design is to leverage on duality of retrosynthesis and reaction prediction. They are a pair of mutual reversible processes that factorize the joint distribution in different orders, where reaction prediction is "forward direction" – $p(y|X)$) and retrosynthesis is the "backward direction" –

$p(X|y)$. With additional prior modeling, the joint probability $p(X, y)$ factorizes to either $p(X|y)p(y)$ or $p(y|X)p(X)$. We propose a training framework that leverages on the duality of the forward and backward directions and performs consistent training between the two to bridge the divergence.

The advantage of the duality of reversible processes has been demonstrated in other applications as well. He et al. [21] trained a reinforcement learning (policy gradient) model to achieve duality in natural language processing and improved performances. Wei et al. [22] treated code summary and code generation as a pair of dual tasks, and improved efficacy by imposing symmetry between attention weights of LSTM encoder-decoder in forward and backward directions. Despite of their encouraging results, these models are not ideal for stable and efficient training for retrosynthesis, as policy gradient methods suffer from high variance and LSTM has sub-optimal performance. Therefore we propose a novel training method that is simple yet efficient for retrosynthesis task. We impose dual-

---

**Algorithm 2** Dual Model

**[Train Phase]: Learning**:
**Input:** Reactants X and product y.
Let $\theta = \{\gamma, \alpha, \eta\}$
Define $E_\theta$ as Eq (7)
$E_\theta = \log p_\gamma(X) + \log p_\alpha(y|X) + \log p_\eta(X|y)$
**1. Train backward**:
$\eta^* = \arg\min_\eta L_{dual} = \arg\max_\eta \widehat{\mathbb{E}}[\log p_\eta(X|y)]$
**2. Train prior and forward**: Plug in $\eta^*$
$p^{mix}(X, y) = \frac{1}{1+\beta}\hat{p}(X, y) + \frac{\beta}{1+\beta}\hat{p}(y)p_{\eta^*}(X|y)$
$\gamma^*, \alpha^* = \arg\min_{\gamma, \alpha} L_{dual}$
$\qquad = \widehat{\mathbb{E}}_{p^{mix}_{(X,y)}}[\log p_\gamma(X) + \log p_\alpha(y|X)]$
**[Test Phase]: Inference**:
**Input**: $\theta^* = \{\gamma^*, \alpha^*, \eta^*\}$, $y^{test}$, Proposal P.
$L \leftarrow P(y^{test})$
$X^* = \arg\min_{X \in L} E_{\theta^*}(X, y^{test})$
**Return** $X^*$

---

ity constraints by training forward direction on a mixture of samples drawn from the backward and original dataset. To our best knowledge, we are the first to apply duality to retrosynthesis and to impose duality constraints by samples drawn from one direction. The EBM is defined:

$$p(X|y) \propto \exp\left(\log p_\gamma(X) + \log p_\alpha(y|X) + \log p_\eta(X|y)\right) \tag{6}$$

$$= \exp(-E_\theta(X, y)) \tag{7}$$

where prior $p(X)$, forward likelihood $p(y|X)$, and backward posterior $P(X|y)$ are modeled as autoregressive models (Sec 2.1.2), parameterized by transformers with parameters $\gamma$, $\alpha$, and $\eta$. Note energy function can be designed free of choice.

The consistent training is achieved by minimizing the "dual loss", where the duality constraints in the equation below are imposed to penalize KL divergence of the two directions, *i.e.*, $\mathrm{KL}(\mathrm{backward}|\mathrm{forward})$. For simplicity, we fix the backward probability in the dual loss, and therefore entropy $H(\mathrm{backward})$ is dropped.

$$\gamma^*, \alpha^*, \eta^* = \arg\min_{\gamma, \alpha, \eta} \ell_{\mathrm{dual}} \tag{8}$$

$$\ell_{\mathrm{dual}} = -\left(\underbrace{\widehat{\mathbb{E}}[\log p_\gamma(X) + \log p_\alpha(y|X)]}_{\text{forward direction}}\right. \tag{9}$$

$$\left. + \underbrace{\beta\widehat{\mathbb{E}}_y\widehat{\mathbb{E}}_{X|y}[\log p_\gamma(X) + \log p_\alpha(y|X)]}_{\text{duality constraints}} + \underbrace{\widehat{\mathbb{E}}[\log p_\eta(X|y)]}_{\text{backward direction}}\right) \tag{10}$$

$$= -\widehat{\mathbb{E}}_{p^{\mathrm{mix}}_{(X,y)}}[\log p_\gamma(X) + \log p_\alpha(y|X)] - \widehat{\mathbb{E}}[\log p_\eta(X|y)] \tag{11}$$

where $\widehat{E}$ indicates expectation over empirical data distribution $\hat{p}(X, y)$. The duality constraints $\beta\widehat{E}_y\widehat{E}_{X|y}[\log p_\gamma(X) + \log p_\alpha(y|X)]$ is the expectation of the forward direction $\log p_\gamma(X) + \log p_\alpha(y|X)$ with respect to empirical backward data distribution $\hat{E}_y\hat{E}_{X|y}$, where $\hat{E}_y\hat{E}_{X|y}$ are approximated by samples drawn from $p_\eta(X|y)$, as y is given so $p(y) = 1$. $\beta$ is scale parameter. In our implementation we use size $k$-beam search to draw samples efficiently. Combining "forward" and "duality constraints" terms (Eq 11), we can see that the first term of the dual loss is to train the forward direction on the mixture distribution of the original data and samples drawn from backward directions $p^{\mathrm{mix}}(X, y) = \frac{1}{1+\beta}\hat{p}(X, y) + \frac{\beta}{1+\beta}\hat{p}(y)p_\eta(X|y)$. Put every piece together (Algorithm 2

and Fig 4 in Appendix). Here is our training procedure. Since we parameterize the three probabilities separately, the optimization of dual loss breaks into two steps:

- **Step 1: Train backward**. $\eta$ does not depend on forward direction under empirical data distribution. $\eta^* = \arg\min_\eta L_{\text{dual}} = \arg\max_\eta \widehat{E}[\log p_\eta(X|y)]$. $\eta$ can be learned by MLE.

- **Step 2: Train prior and forward**. We plug $\eta^*$ into $p_{\eta^*}^{\text{mix}}(X,y)$. $\gamma^*, \alpha^* = \arg\min_{\gamma,\alpha} L_{\text{dual}} = \arg\max_{\gamma,\alpha} \widehat{\mathbb{E}}_{p_{\eta^*}^{\text{mix}}(X,y)}[\log p_\gamma(X) + \log p_\alpha(y|X)]$. $\gamma, \alpha$ can be learned by MLE.

  We provide ablation study of each component of dual loss in Appendix 5.4. The results show that each component in dual loss contribute to the final performance positively.

### 2.1.4 Perturbed model

In contrast to the ordered model that factorizes the sequence in one direction, we use a perturbed sequential model to achieve stochastic bidirectional factorization adapted from XLNet [23]. In particular, this model permutes the factorization order (while maintaining position encoding of the original order) that is used in the forward autoregressive model.

$$p(X|y,z) = p(X^{(z_1)}, X^{(z_2)}, \ldots, X^{(z_{|s(X)|})}|y) = \prod_{i=1}^{|s(X)|} p_\theta(X^{(z_i)}|X^{(z_1:z_{i-1})}, y) \qquad (12)$$

where the permutation order $z$ is a permutation of the original order sequence $z_o = [1, 2, \ldots, |X|]$ and $z_i$ denotes the $i$-th element of permutation $z$. Here $z$ is treated as hidden variable.

### 2.1.5 Bidirectional model

An alternative way to achieve bidirectional context conditioning is the denoising auto-encoding model. We adapt bidirectional model from BERT [24] to our application. The conditional probability $p(X|y)$ is factorized into product of conditional distributions of one random variable conditioning on others,

$$p(X|y) \approx \exp\left( \sum_{i=1}^{|s(X)|} \log p_\theta(X^{(i)}|X^{\neg i}, y) \right) \qquad (13)$$

As presented in Wang and Cho [25], although the model is similar to MRF [26], the marginal of each dimension in Eq (13) does not have a simple form as in BERT training objective. It may result in a mismatch between the model and the learning objective. This model can be trained by pseudo-likelihood (Sec 3.2)

## 2.2 Graph-based Model

Compared with the sequence-based model, the graph-based methods present chemical molecules, with vertices as atoms and edges as chemical bonds. This natural parameterization allows straightforward application of chemistry knowledge by sub-graph matching with templates or reaction centers. We instantiated three representative gragh-based approaches, namely NeuralSym [27], GLN [28] and G2G [15], from the framework. Firstly, we introduce an important concept *template*, which can assist modeling, learning, and inference.

***Templates*** are reaction rules extracted from existing reactions. They are formed by reaction centers (a set of atoms changed, e.g. to form or break bonds). A template $T$ consists of a product-subgraph pattern $(t_y)$ and reactants-subgraph pattern(s) $(t_X)$, denoted as $T := t_y \to t_X$, where $X$ is a molecular set. We overload the notation to define a *template operator* $T(\cdot) : M \mapsto \mathscr{P}(M)$ which takes a product as input, and returns a set of candidate reactant sets. $T(\cdot)$ works as follows: enumerate all the templates with product-subgraph $t_y$ matching with the given product $y$ and define $S(y) = \{T : t_y \in y, \forall T \in \mathcal{T}\}$, where $\mathcal{T}$ are available templates; then reconstruct the reactant candidates by instantiating reactant-subgraphs of the matched templates $R = \{X : t_X \in X, \forall T \in S(y)\}$. The output of $T(\cdot)$ is $R$. $T(\cdot)$ can be implemented by chemistry toolbox RDKit [29].

### 2.2.1 Template prediction: NeuralSym
NeuralSym is a template-based method, which treats the template prediction as multi-class classification. The corresponding probability model under the EBM framework can be written as:

$$p(X|y) \propto \sum_{T \in \mathcal{T}} \exp(e_T^\top f(y)) \mathbb{I}\left[X \in T(y)\right] \tag{14}$$

where $f(\cdot)$ is a neural network that embeds molecule graph $y$, and $e_T$ is the embedding of template $T$. Learning such model requires only optimizing the cross entropy, despite that the number of potential templates could be very large.

### 2.2.2 Graph-matching with template: GLN

Dai et al. [28] proposed a method of graph matching the reactants and products with their corresponding components in the template to model the reactants and template jointly, with the model:

$$p(X, T|y) \propto \exp(w_1(T, y) + w_2(X, T, y)) \cdot \phi_y(T) \phi_{y,T}(X) \tag{15}$$

where $w_1$ and $w_2$ are graph matching score functions, and the $\phi(\cdot)$ operators defines the hard template matching results. This model assigns zero probability to the reactions that do not match with the template. $p(X|y)$ can be obtained by marginalizing over all templates.

### 2.2.3 Graph matching with reaction centers, G2G and GraphRETRO

In contrast with GLN, a few recent works G2Gs [15] and GraphRETRO [17] proposed to predict reaction center directly. These methods closely imitate chemistry experts when performing retrosynthesis: first identify reaction centers (i.e. where the bond breaks, denoted as $c$), then reconstruct $X$.

$$p(X|y) \propto \exp\left(\log\left(\sum_{c \in y} p(X|c, y)p(c|y)\right)\right) \tag{16}$$

All the methods mentioned above require the additional atom-mapping as supervision during training, while NeuralSym and GLN require template information during inference. So NeuralSym and GLN are template-based methods. Since atom mapping plus reaction centers have almost same information as templates, we denote G2G and GraphRETRO method as semi-template-based approach.

## 3 Learning

Training EBMs is to learn parameters $\theta$. In particular, we introduce three ways to learn exact (if applicable) or approximate maximum likelihood estimation (MLE) for full energy-based model (Sec 2.1.1), as this model includes other sequence-based EBM variants (ordered, perturbed, bidirectional, etc) by instantiating $E_\theta$ accordingly. Training EBMs with MLE is non-trivial because the partition function $Z(\theta)$ in Eq (1) is generally intractable. Computing $Z(\theta)$ involves approximation or additional information.

### 3.1 Approximate MLE: integration using template.

We use additional chemistry information: Templates. Direct MLE is not feasible because the partition function of Eq (3) involves enumerating full molecular set $M$, which is intractable. Here we use templates to get a finite support of the partition function. Specifically, we use template operator to extract a set of reactant candidates associated with $y$, denoted as $T(y)$. As the size of $T(y)$ is about tens to hundreds (not computationally prohibitive), we can perform exact inference of Eq (3) to obtain the MLE. We denote this training scheme as *template learning*.

### 3.2 Approximate MLE: pseudo-likelihood.

Alternatively, we can provide an approximation of Eq (3) via pseudo-likelihood [30] to enable training. Pseudo-likelihood factorizes the joint distribution into the product of conditional probabilities of each variable given the rest. Theoretically, the pseudo-likelihood estimator yields an exact solution if the data is generated by a model $p(X|y)$ and number of data points $n \to \infty$ (i.e., it is consistent) [30]. For the full model, training is performed as:

$$p(X|y) \approx \exp\left(\sum_{i=1}^{|s(X)|} \log p_\theta(X^{(i)}|X^{\neg i}, y)\right) = \exp\left(\sum_{i}^{|s(X)|} \log \frac{\exp\left(g_\theta(X, y)\right)}{\sum_{c \in S} \exp\left(g_\theta(X', y; X'^{\neg i} = X^{\neg i}, X'^{(i)} = c)\right)}\right) \tag{17}$$

where the superscript $\neg$ indicates sequence except the $i$-th token and $g_\theta(p, q) : S^{|p|} \times S^{|q|} \mapsto \mathbb{R}$ is a transformer architecture that maps two sequences to a scalar. As bidirectional model Sec 2.1.5

and training approaches Sec 3.2 (approximate joint probability) factorizes in the same way, pseudo-likelihood is a convenient way to train this model.

### 3.3 Exact MLE: tractable factorization.

This training procedure works for a special case of the full model, which has a tractable factorization of the joint probability, e.g., autoregressive models in ordered (Sec 2.1.2) and perturbed (Sec 2.1.4).

### 3.4 Generalized sequence model.

Generalized sequence model first infer latent variable $S^* = \arg\max p(S|y)$ and then infer $X^* = \arg\max p(X|y, S^*)$ as the vanilla sequence-based model with $S$ provided as additional input (e.g. concatenate $S^*$ and $y$).

## 4 Inference

With the trained $E_{\theta^*}$, inference identifies the best $X$ that minimizes the energy function for given $y^{\text{test}}$, i.e. $X^{\text{test}} = \arg\min_{X \in \mathcal{X}} E_{\theta^*}(X, y^{\text{test}})$. Directly solving the above minimization is again intractable, but the energy function can generally be used for ranking. Let $R$ denote the rank of candidate $X_i$ for the given $y^{\text{test}}$ (lower is better).

$$\{R(X_1) < R(X_2) \iff E_{\theta^*}(X_1, y^{\text{test}}) < E_{\theta^*}(X_2, y^{\text{test}})\} \tag{18}$$

Practically, as illustrated in Fig 2, one can use either template-based or template-free method to come up with initial proposals for ranking, as follows.

**Template-based Proposing (TB).** Templates can be used to extract a list of proposed reactant candidates by using templates. We use template operator $T(\cdot)$ (defined in Sec 2.2) to propose a list of candidate reactant sets from the input product $y$. **Template-free Proposing (TF).** In this paper, *template-free ranking* makes proposals using the learned prediction model. We use a simple autoregressive form for $p(X|y)$ (Ordered model), which can draw the top $K$ most likely samples from this distribution using beam search, which is computational efficient.

## 5 Experiments

### 5.1 Experiment setup

Dataset and evaluation used follow existing work [31, 28, 13, 15]. We evaluate our method on a benchmark dataset named USPTO-50k, which includes 50k reactions falling into ten reaction types from the US patent literature. The datasets are split into train/validation/test with percentage of $80\%/10\%/10\%$. Our evaluation metric is the top-$k$ exact match accuracy, referring to the percentage of examples where the ground truth reactant set was found within the top $k$ predictions made by the model. Following the common practice, we use RDKit [29] to canonicalize the SMILES string. For sequence-based models, we incorporate the augmentation trick to ensure best performance. The procedures are as follows: (1) Replace each molecule in reactant set or product using random SMILES; (2) Random permute the order of reactant molecules. The augmentated SMILES are different linearizations of the same molecules. It can prevent sequence-based models (transformer) from over-fitting. However, the augmentation does not improve performance for graph-based models, as graph-based models take graph format as input which is invariant for different augmentations.

### 5.2 Existing methods

We evaluate of our approach against several existing methods, including both template-based, semi-template based and template-free approaches. **Template-free methods**: `Transformer` [14] is a transformer based approach that trains a second transformer to identify the wrong translations and remove them. `LSTM` [13] is a sequence to sequence approach that use LSTM as encoder and decoder. **Template-based methods:** `retrosim` [31] selects template for target molecules using fingerprint based similarity measure between targets and templates; `neuralsym` [27] performs selection of templates as a multiple-class problem using MLP; `GLN` builds a template induced graphical model and makes prediction with approximated MAP.

**Semi-Template based methods**: `G2Gs` [15], `GRetroXpert` [16], and `GraphRETRO` [17] share the same idea: infer reaction center to generate synthons, and then complete the missing pieces (aka "leaving groups") in synthons to generate reactants. These methods use "reaction centers" as additional information to supervise their algorithm. The reaction centers preserve key information in templates. So we denote them as "Semi-Template".

## 5.3 Template-free evaluation

Table 1 shows our best EBM variant (the dual model) evaluated in a template free setup. We first perform evaluations on all the EBM variants introduced in Sec 2.1 to select the best EBM variant. The results show that the dual model outperforms other EBM variants by a clear margin (Table 4 in Appendix). The evaluation is on template-based proposing to ensure the proposal list of candidate molecules is the same for all the variants.

Then we pursue further on template-free setup. An ideal model requires a proposal model with good coverage and a ranking model with good accuracy. We explored various combinations of proposal-ranking pairs. The proposal model evaluated is the ordered model trained on USPTO50K and augmented USPTO50K, respectively. The ranking model is the dual model trained on augmented data, as it performs the best in Table 4. Our best performer is ordered-proposal (USPTO 50K)-dual-ranking (aug USPTO 50K) model. A case study showing how dual model improves accuracy upon proposal is given in Fig 3, where it shows how the energy based re-ranking refines the initial proposal. One interesting observation is that, the proposal ordered model trained on augmented data has higher top 1 accuracy but much lower top 10 accuracy, than the one trained without augmentation. This indicates that the proposal using augmented data has low coverage in the prediction space. We observed that the model learned on augmented dataset learns various representations of the same molecule (due to usage of random SMILES). A certain percentage of proposed candidates are the same after canonicalization, which is good for top 1 prediction during ranking but undesired for proposal.

Table 1: **Template-free: Dual model**: Translation Proposal and Dual Ranking

| Type | Proposal | | | | | | Re-rank | | | | |
|---|---|---|---|---|---|---|---|---|---|---|---|
| | Proposal model | Top 1 | Top 5 | Top 10 | Top 50 | Top 100 | Rank model | Top 1 | Top 3 | Top 5 | Top 10 |
| No | Ordered on UPSPTO | 44.4 | 64.9 | 69.9 | 77.2 | 78.0 | Dual trained on Aug USPTO | 53.6 | **70.7** | **74.6** | **77.0** |
| | Ordered on Aug USPTO | 53.2 | 54.7 | 55.6 | 60.5 | 60.5 | | **54.5** | 60.0 | 60.4 | 60.5 |
| | - | - | - | - | - | - | SOTA (RetroXpert [16]) | 50.4 | 61.1 | 62.3 | 63.4 |
| Yes | Ordered on USPTO | 56.0 | 76.1 | 79.7 | 85.2 | 86.4 | Dual trained on Aug USPTO | 65.7 | **81.9** | **84.7** | **85.9** |
| | Ordered on Aug USPTO | 64.7 | 66.5 | 67.3 | 69.7 | 75.7 | | **66.2** | 75.1 | 75.6 | 75.7 |
| | - | - | - | - | - | - | SOTA (RetroXpert [16]) | 62.1 | 75.8 | 78.5 | 80.9 |

## 5.4 Ablation Study of the dual loss

Since the dual variant serves as the backbone variant in the previous section, we perform additional ablation study to investigate the performance of the dual variant with respect to different designs of the dual loss. Table 2 shows that each component of the dual loss contribute positively to the final performance. The dual constraint leads to additional improvement on the top of other components, which is more challenging to achieve in a higher accuracy region.

The evaluation of Table 2 is under the same setup as Table 4 – uses template-based proposal for fair and easy comparison. The notations of Table 2 are as follows: The "dual" row are entries taken from Table 4, showing results trained with dual loss. To recap, the dual loss is defined in Eq (10) and the dual constraint is its middle term. $\widehat{\mathbb{E}}[\log p_\gamma(X) + \log p_\alpha(y|X) + \log p_\eta(X|y)]$ is the dual loss without the dual constraint. $\widehat{\mathbb{E}}[\log p_\alpha(y|X) + \log p_\eta(X|y)]$ is the dual loss without the prior $\log p_\gamma(X)$. $\log p_\eta(X|y)]$ is only including backward direction.

Table 2: **Ablation Study of dual loss design when reaction type is known**

| Aug USPTO | Top 1 | Top 3 | Top 5 | Top 10 |
|---|---|---|---|---|
| Dual | **67.7** | **84.8** | **88.9** | **92.0** |
| $\widehat{\mathbb{E}}[\log p_\gamma(X) + \log p_\alpha(y|X) + \log p_\eta(X|y)]$ | 67.0 | 84.7 | 88.9 | 91.95 |
| $\widehat{\mathbb{E}}[\log p_\alpha(y|X) + \log p_\eta(X|y)]$ | 66.1 | 82.8 | 87.6 | 91.3 |
| $\widehat{\mathbb{E}}[\log p_\eta(X|y)]$ | 60.9 | 80.9 | 85.8 | 90.2 |

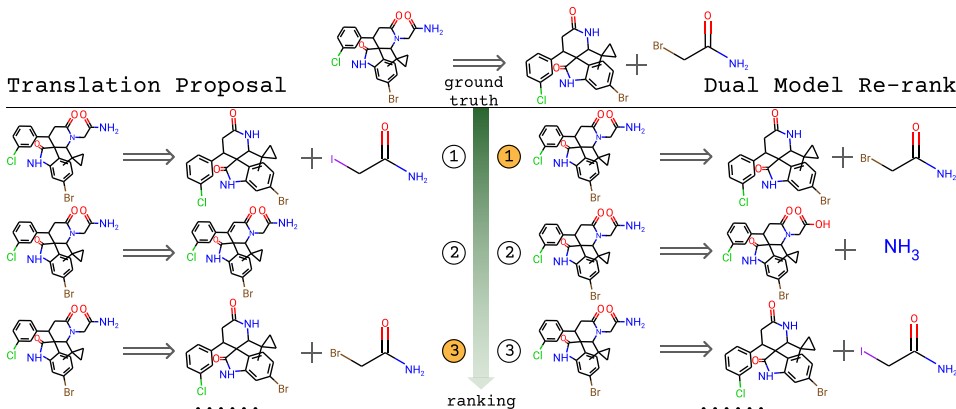

Figure 3: **Dual ranking improves upon translation proposal.** Left and right column are the top three candidates from translation proposal and dual re-ranking of the proposal. Ground truth (GT) is given at the top and is labeled orange in the middle. By dual re-ranking, the GT ranks the first place, whereas the 3rd place in the proposal. Note that the first place in the proposal is only one atom different from GT (Br vs I), indicating the dual model is able to identify small changes in structure. Another example is given in Fig 5 in Appendix.

Table 3: **Top K exact match accuracy** of existing methods

| Category | Model | Reaction type unknown | | | | Reaction type known | | | |
|---|---|---|---|---|---|---|---|---|---|
| | | top1 | top3 | top5 | top10 | top1 | top3 | top5 | top10 |
| TB | retrosim [31] | 37.3 | 54.7 | 63.3 | 74.1 | 52.9 | 73.8 | 81.2 | 88.1 |
| | NeuralSym [27] | 44.4 | 65.3 | 72.4 | 78.9 | 55.3 | 76.0 | 81.4 | 85.1 |
| | GLN [28] | 52.5 | 69.0 | 75.6 | 83.7 | 64.2 | 79.1 | 85.2 | 90.0 |
| Semi-TB | G2Gs [15] | 48.9 | 67.6 | 72.5 | 75.5 | 61.0 | 81.3 | 86.0 | 88.7 |
| | GraphRETRO [17] | 53.7 | 68.3 | 72.2 | 75.5 | 63.9 | 81.5 | 85.2 | 88.1 |
| | RetroXpert [16] | 50.4 | 61.1 | 62.3 | 63.4 | 62.1 | 75.8 | 78.5 | 80.9 |
| TF | LSTM [13] | - | - | - | - | 37.4 | 52.4 | 57.0 | 61.7 |
| | Transformer [14] | 43.7 | 60.0 | 65.2 | 68.7 | 59.0 | 74.8 | 78.1 | 81.1 |
| | Dual (Ours) | **53.6** | **70.7** | **74.6** | **77.0** | **65.7** | **81.9** | **84.7** | **85.9** |

## 5.5 Comparison against the state-of-the-art

Table 3 presents the main results. All the baseline results are extracted from existing works as we share the same experiment protocol. The dual model is trained with randomized SMILES to inject order invariance information of molecule graph traversal. Note that other methods like graph-based variants do not require such randomization as the graph representation is already order invariant. We can see that, regarding top 1 accuracy, our proposed dual model outperforms the current state-of-the-art methods. Semi-template methods are those require ground truth reaction centers during training as supervision, whereas generalized dual model does not require this additional information, yet still output perform the best semi-template models by $1.6\%$ when reaction type is known. This demonstrates the advantages of the dual model. RetroXpert results are the updated results taken from `https://github.com/uta-smile/RetroXpert`.

## 6 Conclusion

In this paper we proposed an unified EBM framework that integrates multiple sequence- and graph-based variants for retrosynthesis. Assisted by a comprehensive assessment, we provide a critical understanding of different designs. Based on this, we proposed a novel variant – generalized dual model, which outperforms state-of-the-art in template free manner.

