# References

[1] EJ Corey. Robert robinson lecture. retrosynthetic thinking—essentials and examples. *Chemical Society Reviews*, 17:111–133, 1988.

[2] Elias James Corey. The logic of chemical synthesis: multistep synthesis of complex carbogenic molecules (nobel lecture). *Angewandte Chemie International Edition in English*, 30(5):455–465, 1991.

[3] Marwin HS Segler, Mike Preuss, and Mark P Waller. Planning chemical syntheses with deep neural networks and symbolic ai. *Nature*, 555(7698):604–610, 2018.

[4] Sara Szymkuć, Ewa P Gajewska, Tomasz Klucznik, Karol Molga, Piotr Dittwald, Michał Startek, Michał Bajczyk, and Bartosz A Grzybowski. Computer-assisted synthetic planning: The end of the beginning. *Angewandte Chemie International Edition*, 55(20):5904–5937, 2016.

[5] Felix Strieth-Kalthoff, Frederik Sandfort, Marwin HS Segler, and Frank Glorius. Machine learning the ropes: principles, applications and directions in synthetic chemistry. *Chemical Society Reviews*, 49(17):6154–6168, 2020.

[6] Connor W Coley, Regina Barzilay, Tommi S Jaakkola, William H Green, and Klavs F Jensen. Prediction of organic reaction outcomes using machine learning. *ACS central science*, 3(5): 434–443, 2017.

[7] Connor W Coley, Dale A Thomas, Justin AM Lummiss, Jonathan N Jaworski, Christopher P Breen, Victor Schultz, Travis Hart, Joshua S Fishman, Luke Rogers, Hanyu Gao, et al. A robotic platform for flow synthesis of organic compounds informed by ai planning. *Science*, 365(6453): eaax1566, 2019.

[8] Marwin HS Segler and Mark P Waller. Modelling chemical reasoning to predict and invent reactions. *Chemistry–A European Journal*, 23(25):6118–6128, 2017.

[9] Simon Johansson, Amol Thakkar, Thierry Kogej, Esben Bjerrum, Samuel Genheden, Tomas Bastys, Christos Kannas, Alexander Schliep, Hongming Chen, and Ola Engkvist. Ai-assisted synthesis prediction. *Drug Discovery Today: Technologies*, 2020.

[10] David Weininger. Smiles, a chemical language and information system. 1. introduction to methodology and encoding rules. *Journal of chemical information and computer sciences*, 28 (1):31–36, 1988.

[11] Ashish Vaswani, Noam Shazeer, Niki Parmar, Jakob Uszkoreit, Llion Jones, Aidan N Gomez, Łukasz Kaiser, and Illia Polosukhin. Attention is all you need. In *Advances in neural information processing systems*, pages 5998–6008, 2017.

[12] Philippe Schwaller, Teodoro Laino, Théophile Gaudin, Peter Bolgar, Christopher A Hunter, Costas Bekas, and Alpha A Lee. Molecular transformer: A model for uncertainty-calibrated chemical reaction prediction. *ACS central science*, 5(9):1572–1583, 2019.

[13] Bowen Liu, Bharath Ramsundar, Prasad Kawthekar, Jade Shi, Joseph Gomes, Quang Luu Nguyen, Stephen Ho, Jack Sloane, Paul Wender, and Vijay Pande. Retrosynthetic re- action prediction using neural sequence-to-sequence models. *ACS central science*, 3(10): 1103–1113, 2017.

[14] Shuangjia Zheng, Jiahua Rao, Zhongyue Zhang, Jun Xu, and Yuedong Yang. Predicting retrosynthetic reactions using self-corrected transformer neural networks. *Journal of Chemical Information and Modeling*, 2019.

[15] Chence Shi, Minkai Xu, Hongyu Guo, Ming Zhang, and Jian Tang. A graph to graphs framework for retrosynthesis prediction. *arXiv preprint arXiv:2003.12725*, 2020.

[16] Chaochao Yan, Qianggang Ding, Peilin Zhao, Shuangjia Zheng, Jinyu Yang, Yang Yu, and Junzhou Huang. Retroxpert: Decompose retrosynthesis prediction like a chemist. 2020.

[17] Vignesh Ram Somnath, Charlotte Bunne, Connor W Coley, Andreas Krause, and Regina Barzilay. Learning graph models for retrosynthesis prediction. *arXiv preprint arXiv:2006.07038*, 2020.

[18] Yann LeCun, Sumit Chopra, Raia Hadsell, M Ranzato, and F Huang. A tutorial on energy-based learning. *Predicting structured data*, 1(0), 2006.

[19] Geoffrey E Hinton. A practical guide to training restricted boltzmann machines. In *Neural networks: Tricks of the trade*, pages 599–619. Springer, 2012.

[20] Ilya Sutskever, Oriol Vinyals, and Quoc V Le. Sequence to sequence learning with neural networks. In *Advances in neural information processing systems*, pages 3104–3112, 2014.

[21] Di He, Yingce Xia, Tao Qin, Liwei Wang, Nenghai Yu, Tie-Yan Liu, and Wei-Ying Ma. Dual learning for machine translation. In *Advances in neural information processing systems*, pages 820–828, 2016.

[22] Bolin Wei, Ge Li, Xin Xia, Zhiyi Fu, and Zhi Jin. Code generation as a dual task of code summarization. In *Advances in Neural Information Processing Systems*, pages 6559–6569, 2019.

[23] Zhilin Yang, Zihang Dai, Yiming Yang, Jaime Carbonell, Russ R Salakhutdinov, and Quoc V Le. Xlnet: Generalized autoregressive pretraining for language understanding. In *Advances in neural information processing systems*, pages 5754–5764, 2019.

[24] Jacob Devlin, Ming-Wei Chang, Kenton Lee, and Kristina Toutanova. Bert: Pre-training of deep bidirectional transformers for language understanding. *arXiv preprint arXiv:1810.04805*, 2018.

[25] Alex Wang and Kyunghyun Cho. Bert has a mouth, and it must speak: Bert as a markov random field language model. *arXiv preprint arXiv:1902.04094*, 2019.

[26] Ross Kindermann. Markov random fields and their applications. *American mathematical society*, 1980.

[27] Marwin HS Segler and Mark P Waller. Neural-symbolic machine learning for retrosynthesis and reaction prediction. *Chemistry–A European Journal*, 23(25):5966–5971, 2017.

[28] Hanjun Dai, Chengtao Li, Connor Coley, Bo Dai, and Le Song. Retrosynthesis prediction with conditional graph logic network. In *Advances in Neural Information Processing Systems*, pages 8870–8880, 2019.

[29] G Landrum. Rdkit: Open-source cheminformatics software, 2016.

[30] Julian Besag. Statistical analysis of non-lattice data. *Journal of the Royal Statistical Society: Series D (The Statistician)*, 24(3):179–195, 1975.

[31] Connor W Coley, Luke Rogers, William H Green, and Klavs F Jensen. Computer-assisted retrosynthesis based on molecular similarity. *ACS central science*, 3(12):1237–1245, 2017.

[32] Andrew Dalke. Deepsmiles: An adaptation of smiles for use in. 2018.

[33] Mario Krenn, Florian Häse, AkshatKumar Nigam, Pascal Friederich, and Alán Aspuru-Guzik. Selfies: a robust representation of semantically constrained graphs with an example application in chemistry. *arXiv preprint arXiv:1905.13741*, 2019.

[34] Guillaume Klein, Yoon Kim, Yuntian Deng, Jean Senellart, and Alexander M Rush. Opennmt: Open-source toolkit for neural machine translation. *arXiv preprint arXiv:1701.02810*, 2017.

[35] Diederik P Kingma and Jimmy Ba. Adam: A method for stochastic optimization. *arXiv preprint arXiv:1412.6980*, 2014.

# A Appendix

## A.1 Terminology: Reaction center and Synthons

Reaction center of a chemical reaction are the bonds that are broken or formed during a chemical reaction. For retrosynthesis, reaction centers are bonds exist in product, but do not exist in reactants. One chemical reaction may have multiple reaction centers. Synthons are the sub-parts extracted from the products by breaking the bonds in the reaction center. Synthons are usually not valid molecules with $*$ to indicate the broken ends in the reaction centers.

## A.2 Sequence-based variant evaluation

In this section, we mainly compare different energy based sequence models described in Sec 2.1. Table 4 provides the results of each sequence model variant described in Sec 2.1. For simplicity of the proposal, we evaluate them using *template-based ranking* described in Sec 4. Each variant is evaluated on USPTO 50K and augmented USPTO 50K using random SMILES. Without reiterating good performance for the dual variant, we focus on discussion of variants with undesired performance. The perturbed sequential model (Sec 2.1.4) and bidirectional model (Sec 2.1.5) are inferior to dual or ordered models, where the main reason possibly comes from the fact that the learning objective approximates the actual model and Eq (13) poorly, and thus leads to discrepancy between training and inference. The full model (Sec 2.1.1) despite being most flexible and achieving best top 10 performance when type is given, would suffer from high computation cost due to the explicit integration even with the templates. In addition to the understanding of individual models throughout the comprehensive study, we find it is important to balance the trade-off between model capacity and learning tractability. A powerful model without effective training would be even inferior to some well trained simple models. Our dual model makes a good balance between capacity and learning tractability.

Table 4: **Ablation study: Top K accuracy of sequence variants**

| Models | \multicolumn{4}{c}{Reaction type unknown} | | | | \multicolumn{4}{c}{Reaction type known} | | | |
| --- | --- | --- | --- | --- | --- | --- | --- | --- |
| | Top 1 | Top 3 | Top 5 | Top 10 | Top 1 | Top 3 | Top 5 | Top 10 |
| Ordered | 54.2 | 72.0 | 77.7 | 84.2 | 66.4 | 82.9 | 87.4 | 91.0 |
| Perturbed | 47.3 | 64.6 | 70.4 | 75.8 | 64.2 | 79.8 | 83.3 | 86.4 |
| Bidirectional | 23.5 | 43.7 | 54.3 | 69.5 | 41.9 | 66.3 | 75.6 | 84.6 |
| Dual | **55.2** | **74.6** | **80.5** | **86.9** | **67.7** | **84.8** | **88.9** | **92.0** |

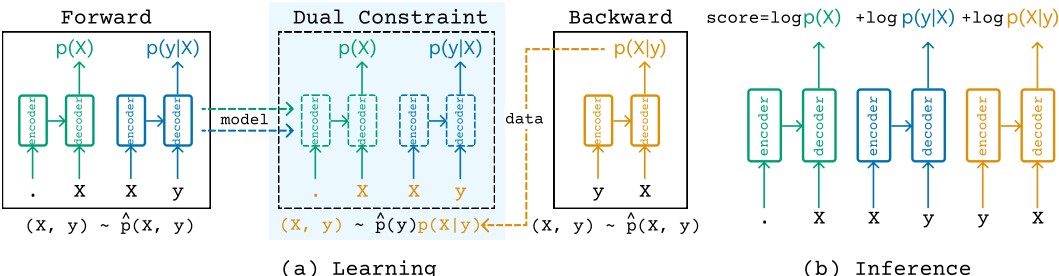

Figure 4: **Dual model. (a) Learning** consists of training three transformers: prior $p(X)$ (green), likelihood $p(y|X)$ (blue), and backward $p(X|y)$ (orange). Dual model penalizes the divergence between forward $p(X)p(y|X)$ and backward direction $p(y|X)$ with Dual constraint (highlighted). **(b) Inference** Given reactant candidates list, we rank them using Eq (7).

## A.3 Time and space complexity analysis

In this section, we provide time and space complexity regarding model design choices. As the main bottleneck is the computation of transformer model, we measure the complexity in the unit

of transformer model calls. For all the models, the inference only requires the evaluation of (un-normalized) score function, thus the complexity is $O(1)$; For training, the methods that factored have an easy form of likelihood computation, where a diagonal mask is applied to input sequence so that autoregressive is done in parallel (not $|s(x)|$ times), so it requires $O(1)$ model calls. This include ordered/perturbed/bidirectional/dual models. For the full model trained with pseudo-likelihood, it requires $O(|X| \cdot |S|)$ calls due to the evaluation per each dimension and character in vocabulary. Things would be a bit better when trained with template-based method, in which it requires $O(|T(y)|)$ calls, which is proportional to the number of candidates after applying template operator.

As the memory bottleneck is also the transformer model, it has the same order of growth as time complexity with respect to sequence length and vocabulary size. In summary we can see the Full model has much higher cost for training, which might lead to inferior performance. Our dual model with a consistency training objective has the same order of complexity than other autoregressive ones, while yields higher capacity and thus better performance.

### A.4 Example of case study

Here we provide another case study showing with dual model ranking (Sec 2.1.3), the accuracy improves upon translation proposal. Please see Fig 3 and Fig 5.

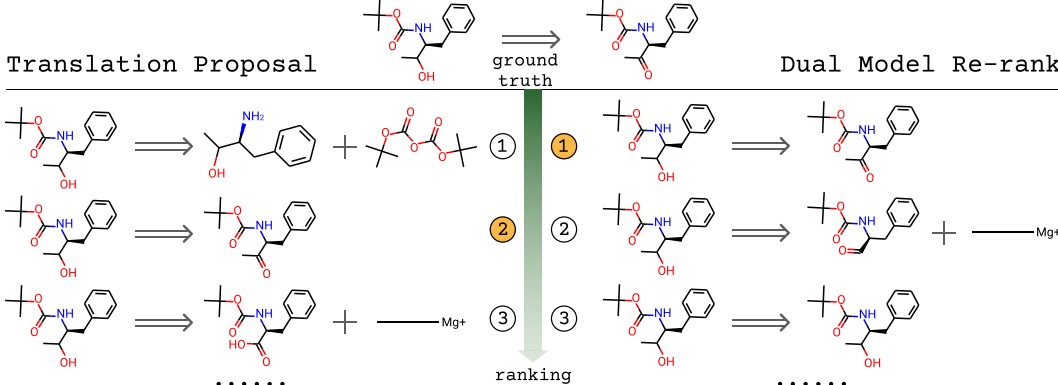

Figure 5: **Dual ranking improves upon translation proposal.** Another example. Descriptions see Fig 3

### A.5 Alternative of SMILES: deepSMILES and SELFIES

In this section, we explore the effect of prepossessing procedure of sequence-based model, e.g. inline representation of molecular graph, in effecting performance of sequence-based model. In particular, deepSMILES [32] and SELFIES [33] are alternatives to SMILES. Without loss of fairness, we evaluated these representations using Ordered sequential model (Sec 2.1.2)The results indicate SMILES work the best. We speculate the reason are deepSMILES and SELFIES are on average longer than SMILES, leading to higher probability of making mistakes on token level and therefore low sequence-level accuracy.

### A.6 Transformer implementation of Permutation Invariant of reactant set

Transformer has a position encoding to mark the different locations on an input sequence. We modified the position encoding such that each molecule starts with 0 encoding, instead of the concatenated position in the reactants sequence. The results are Table 6. We can see that this position encoding is beneficial for non-augment data, but not augment data, as the latter has already considered the permutation invariance order of reactants by data augmentation. In this paper, we use data augmentation to maintain order-invariant for reactants.

Table 5: **deepSMILES and SELFIES**

| Models | Top 1 | Top 3 | Top 5 | Top 10 |
|---|---|---|---|---|
| **SMILES** | | | | |
| Ordered | 47.0 | 67.4 | 75.4 | 83.1 |
| **deepSMILES** | | | | |
| Ordered | 46.08 | 65.87 | 73.54 | 81.51 |
| **Selfies** | | | | |
| Ordered | 43.00 | 62.51 | 70.16 | 79.07 |

Table 6: **Transformer model with permutation invariant position encoding**

| Reaction type is unknown | **USPTO 50k** | | | | |
|---|---|---|---|---|---|
| Models | Top 1 | Top 2 | Top 3 | Top 5 | Top 10 |
| Ordered | 46.97 | 60.71 | 67.39 | 75.35 | 83.14 |
| Ordered + Permutation invariant | 47.29 | 61.29 | 68.08 | 75.37 | 83.36 |
| | **Augmented data** | | | | |
| Ordered | 54.24 | 66.33 | 72.02 | 77.67 | 84.22 |
| Ordered + Permutation invariant | 53.45 | 66.61 | 72.58 | 78.33 | 85.42 |

## A.7 Discussion

**V.1 Full model** (Sec 2.1.1) Full model (Sec 2.1.1) with template learning reaches accuracy of $39.5\%$ and $53.7\%$ on USPTO50k data-sets. Full model is partially limited by expensive computation due to the number of candidates per product.

**V.2 Perturbed sequential model** (Sec 2.1.4) During training, permutation order $z$ is randomly sampled and uses the following training objective:

$$p(X|y) \approx \exp\left(\mathbb{E}_{z \sim Z_{|s(x)|}}\left[\sum_{i=1}^{|X|} \log p_\theta(X^{(z_i)}|z_i, X^{(z_1:z_{i-1})}, y)\right]\right) \tag{19}$$

and the corresponding parameterization:

$$p_\theta(X^{(z_i)}|z_i, X^{(z_1:z_{i-1})}, y) = \log \frac{\exp\left(h(X^{(z_1:z_{i-1})}, z_i, y)^\top e(X^{z_i})\right)}{\sum_{c \in S} \exp\left(h(X^{(z_1:z_{i-1})}, z_i, y)^\top e(c)\right)} \tag{20}$$

where $z_i$ encodes which position index in the permutation order to predict next, implemented by a second position attention (in addition to the primary context attention). Note that Eq (19) is actually a lower bound of the latent variable model, due to Jensen's inequality. However, we focus on this model design for simplicity of permuting order in training.

The lower-bound approximation is tractable for training. Perturbed sequential model has about $\sim 4\%$ accuracy loss in top 1 accuracy compared with ordered model (Sec 2.1.2). We argue the reason are as follows: firstly, we designed $E_\theta$ as the middle term of Eq (19) to facilitate perturbing the order during training, following [23]. However, due to Jensen's inequality, this design is not equal to $P(X|y)$, which causes discrepancy in ranking (inference).

**V.3 Bidirectional model** (Sec 2.1.5) Bidirectional model, however, does not perform well in our experiments. The bidirectional-awareness makes the prediction of one position given all the rest of the sequence $p(X^{(i)}|X^{\neg i}, y)$ almost perfect ($99.9\%$ accuracy in token-level). However, due to

the gap between pseudo-likelihood and maximum likelihood, *i.e.*, $\log P(X|y)$, the performance for predicting the whole sequence will be inferior, as we observed in the experiments.

## A.8    Transformer architecture and training details

The implementation of variants in framework is based on OpenNMT-py [34]. Following [11], transformer is implemented as encoder and decoder, each has a 4 self-attention layers with 8 heads and a feed-forward layer of size 2048. We use model size and word embedding size as 256. Batch size contains 4096 tokens, which approximately contains 20-200 sequences depending on the length of sequence. We trained for 500K steps, where each update uses accumulative gradients of four batches. The optimization uses Adam [35] optimizer with $\beta_1 = 0.9$ and $\beta_2 = 0.998$ with learning rate described in [11] using 8000 warm up steps. The training takes about 48 hours on a single NVIDIA Tesla V100. The setup is true for training transformer-based models, including ordered sequential model (Sec 2.1.2), perturbed sequential model (Sec 2.1.4), bidirectional model (Sec 2.1.5), dual model (Sec 2.1.3). As for full model (Sec 2.1.1), each sample contains 20-500 candidates. We implemented as follows: each batch only contains one sample. Its tens or hundreds of candidates are computed in parallel within the batch. The model parameters are updated when accumulating 100 batches to perform one step of update.