# OpenReview forum: "Towards understanding retrosynthesis by energy-based models"
_NeurIPS.cc/2021/Conference — NeurIPS 2021 Poster_

### Official Review · Reviewer_r7dk · 2021-07-03

**Rating:** 7
**Confidence:** 4

**Summary:**

This paper presents a unified framework with energy-based models for retrosynthesis prediction. Different methods (sequence and graph methods) can be untied with different energy functions within the framework. A dual variant model is presented and it performs consistent forward and backward training. Experimental results demonstrate this consistent training improves the retrosynthesis prediction significantly.


**Limitations And Societal Impact:**

Not

**Main Review:**

The authors present a general EBM framework to unify existing sequence-based and graph-based methods. Through investigating different designs of the EMB framework, a dual variant model is proved to achieve the best performance on USPTO-50K dataset. The proposed dual model also trains a forward reaction prediction model, combined with the retrosynthesis prediction, the top-k predictions can be ranked according to the overall prediction scores. While previous methods rank predictions without incorporating forward-prediction scores.

One major concern for me is that the authors present their methods and variants with abstract formulations, while implementation details are largely ignored. This makes their method difficult to understand. For example, line #101, p(Xi|X1:i-1, y) is parameterized by the Transformer model h(p, q), how does h(p, q) output a logits vector for each value in vocabulary? Does the formula p(Xi|X1:i-1, y) indicate the input is X1:i-1 and y, and output is Xi? Line 142, could the authors add more details to explain how KL divergence is related to the duality constraints?

Another concern for me is the preprocessing of the USPTO-50k dataset. It is found that there is a potential information leak with the dataset reported by the authors of RetroXpert. Have the authors removed the atom mapping and then canonicalized the product SMILES during the inference? Have the authors also run rxnmapper on testing data? It is not necessary to have mapping information for testing data, the authors should remove any mapping information for testing data and then run the inference. Are the authors aware of any potential information leak with the proposed method?

The authors claim that "we propose a unified energy-based model (EBM) framework that integrates sequence- and graph-based models for retrosynthesis." and I do not think this is a significant contribution. EBM is a general framework and it is not surprising that EBM can be used to explain different models, especially the graph-based method GLN itself is energy-based.

The Transformer has been extensively explored for retrosynthesis, the authors did not clarify the difference between their different variants and existing Transformer-based methods. The proposed generalized sequence models at line #216 are similar to the RetroXpert, except that the synthons are directly predicted here. The ordered model at line #98 is similar to the one-stage Transformer methods such as SCROP.

Line #223, previous methods infer reaction center from provided atom-mapping information, which is extracted from third-party software like Indigo. The authors use rxnmapper to find the atom-mapping information. I do not think there is an essential difference.

What is the difference between formula (4) and (13)？

When incorporating templates, what is the difference between the proposed method and GLN except for the sequence and graph-based?

Could the authors also report results on USPTO-full as GLN to demonstrate the scalability?

Line #250, the explanation of the superscript should be moved to line #173 when it first appears.

Line#121, the RHS of the equation in Algorithm 2 should have a minus symbol.




**Time Spent Reviewing:**

12

---

> ### Author Response · Authors · 2021-08-10
> **Thank you**
>
> Thank you very much for your insightful reviews and the time spent on reviewing our paper. We really appreciate your detailed reviews and helpful discussions. Our paper can be improved greatly with your suggestions!
>
>  **Re: For example, line #101, p(Xi|X1:i-1, y) is parameterized by the Transformer model h(p, q), how does h(p, q) output a logits vector for each value in vocabulary? Does the formula p(Xi|X1:i-1, y) indicate the input is X1:i-1 and y, and output is Xi?**
> Thank you for the questions and sorry for the confusion. For each X_i,  h outputs a vector of vocabulary size, then we softmax/cross-entropy loss with true label (the supervision during training). Then the learned output are the logits corresponding to vocabulary. Yes, the input is X{1:i-1} and y, and the output is Xi. However, during training every X_i for all i values in [0, len(X_i)] is simultaneously trained using an upper triangle mask to ensure computational efficiency as it is in "attention is all you need paper"
>
>
> **Re: "Line 142, could the authors add more details to explain how KL divergence is related to the duality constraints?**
> Yes, definitely. KL divergence is to panelize the difference between backward and forward direction. The dual constraints (eq10,   explained in line # 146-149)
>
> KL(P | Q) = H(P, Q) - H(P).
>
> where H is entropy. P is backward distribution p(y)p(X|y); Q is forward p(X)p(y|X).
>  The dual constraints in eq10 is H(P. Q). For simplicity, in duall loss we fix P so H(P) is dropped.
>
>
> **Re: potential information leak**  \
> Please refer to the comment with title "Re "information leak issue reported in RetroXpert": Updated results"
>
>
> **Re: The Transformer has been extensively explored for retrosynthesis, the authors did not clarify the difference between their different variants and existing Transformer-based methods**
>
> Thanks for the comments! Our novelty is that EBM framework that unifies different variants of retrosynthesis algorithms and dual variant is novel for the retrosynthesis setup, especially the dual constraints are achieved by samples drawn from one direction. This approach is simple and stable.
>
>
> **Re: What is the difference between formula (4) and (13)**
> (4) is autoregressive model: Xi conditioned on X1:i-1 \
> (13) is masked model: Xi conditioned on X\i, where X\i is all the tokens but ith token.
>
> **Re: When incorporating templates, what is the difference between the proposed method and GLN except for the sequence and graph-based?** \
> We do not use template during training. In the template-based dual model, we only use templates during testing to propose candidates (for ranking later)., whereas GLN conducts training by matching products or reactants with the templates during training. Our model relies on templates less.
>
>
> **Re:Report results on USPTO-full?**
>  Sure, we will include those results
>
> Thank you very much.

---

> > ### Comment · Reviewer_r7dk · 2021-08-12
> > **Thanks for the respone**
> >
> > Thanks for the authors' detailed response which answered my questions.
> >
> > Please the authors include more implementation details to make the proposed method easier to understand.
> >
> > I decided to increase my score to accept after reading all reviews and responses from the authors.

---

> > > ### Author Response · Authors · 2021-08-13
> > > **Include more implementation details**
> > >
> > > We thank the reviewer for your support on our work and for the time you spent reviewing it. Our paper benefits from your input!
> > >
> > > We apologize for not making the paper easy enough to read.  Yes, absolutely. We will modify the draft to include more implementation details, including implementation of h，KL divergence on dual constraints, and clarifying notations.

---

### Official Review · Reviewer_vxrH · 2021-07-07

**Rating:** 7
**Confidence:** 4

**Summary:**

This paper attempts to describe sequence-based and graph-based one-step retrosynthesis models, which are the current mainstreams of retrosynthesis, in a unified manner with the Energy-Based Model (EBM).
This paper reformulates several existing models with the EBM, mainly focusing on the difference in the energy function definitions.
In addition, a new EBM variant is proposed in this paper, and its performance is updated the State-of-the-Art.

Aside from the mainstreams of the machine learning community, AIzynthfinder recently draws many attention as a free and powerful multi-step retrosynthesis engine for computational chemistry. Some comments on this engine may be beneficial for placing the submitted work in a broader context.

Thakkar A, Kogej T, Reymond J-L, et al (2019) Datasets and their influence on the development of computer assisted synthesis planning tools in the pharmaceutical domain. Chem Sci. https://doi.org/10.1039/C9SC04944D
Genheden S, Thakkar A, Chadimova V, et al (2020) AiZynthFinder: a fast, robust and flexible open-source software for retrosynthetic planning. J. Cheminf. https://jcheminf.biomedcentral.com/articles/10.1186/s13321-020-00472-1


**Ethical Concerns:**

No specific concerns

**Limitations And Societal Impact:**

No specific opinions

**Main Review:**

## Originality:
As far as I know, this paper is the first attempt to describe various Retrosynthesis models in a unified manner in EBM, and it is novel in this respect.
The bi-directional training of sequence models (Dual in this paper) has been shown to be useful in several previous studies, as cited in the text, and is not novel in itself.
However, the use of sampling to stably apply the method to retrosynthesis is a good idea.

## Quality:
I find no evident errors in terms of technical content.

## Clarity:
It is good that the paper carefully introduces the EBM.
On the other hand, since many models have to be covered in one paper, the explanation of each model is limited to a minimum.   I think that the current manuscript requires the reader to have some expertises in retrosynthesis.
In Sec. 2, various existing models are actually formulated by EBM.
In the current manuscript, readers have to go back and forth between pages while remembering the formulas to confirm the differences between the model equations.
If it is possible, it would be easier for readers if the qualitative characteristics of each formula are summarized in one table in the appendix.

## Significance:
Numerical experiments show that a new State-of-the-Art for the one-step retrosynthesis has been achieved, which strongly supports the usefulness of the generalized Dual model proposed in the paper.
In addition, the fact that an unsupervised atom mapping transformer can achieve great performance is good news for readers who have difficulty in handling graph data (in fact, it is not an easy task for many ones).
The Appendix contains many additional experiments and information to evaluate the model from various points of view, which is useful to help the reader understand it.


(+) EBM interpretation of several retrosynthesis models are new
(+) Specific formulation for dual (bi-directional) training of generalized sequence models.
(+) new SotA
(+) appendix informative
(-) cumbersome notations and presentations (partially inevitable because this paper covers many models: understandable)

## after feedbacks
Feedbacks from authors and additional experiment report solves many concerns of reviewers.
I believe this paper is worth publication in NeurIPS.
I keep my positive score.

**Time Spent Reviewing:**

5

---

> ### Author Response · Authors · 2021-08-10
> **Thank you**
>
> Thank you so much for your time and efforts on reviewing our work. Thank you for your support for our article.
> We are happy that you find this paper interesting. We really appreciate your constructive and helpful suggestions. We think modifications based on your suggestions can make our paper better. Thank you.
>
> **Re: placing the submitted work in a broader context: such as multiple steps retrosynthesis planning**
>
> We think the suggested papers are really interesting. One step retrosynthesis is definitely not the complete story of computer assisted synthesis planning. We are also interested in applying our methods to multiple steps planning, especially on AiZynthFinder (Monte Carlo tree search based multiple steps  retrosynthesis planner)
>
> **Re: “In the current manuscript, readers have to go back and forth between pages while remembering the formulas to confirm the differences between the model equations. If it is possible, it would be easier for readers if the qualitative characteristics of each formula are summarized in one table in the appendix.**
>
> We apologize for the inconvenience in reading when you have to go back and forth. Absolutely, that is a great idea! We will include a qualitative characteristics of each formula in the appendix.

---

### Official Review · Reviewer_2KjZ · 2021-07-15

**Rating:** 7
**Confidence:** 5

**Summary:**

The paper summarizes many different models for for retrosynthesis under the EBM framework.
It introduces dual training.
By performing reaction center annotation, state of the art results are claimed.

**Ethical Concerns:**

No concerns, basic research.

**Limitations And Societal Impact:**

No concerns wrt Societal Impact: basic research.

**Main Review:**

The authors unify many recently proposed models for chemical retrosynthesis under the framework of energy based models (EBM)
Furthermore they propose dual training for re-ranking the predictions.
The authors report state of the art performance.

Several works (e.g. Segler 2018, Coley 2019) have combined retro-prediction with forward scoring for re-ranking, so the novelty of this aspect is somewhat limited.

The presentation of the paper is somewhat dense, which mostly due to the length limit.

Overall i think the paper could eventually fit into NEURIPS and is interesting, but there are several open questions which need to be addressed before the publication can be recommended. (see update after rebuttal)


Please note that retroXpert https://github.com/uta-smile/RetroXpert and graphretro https://arxiv.org/abs/2006.07038 results have been updated (they are lower than originally reported). so these results should be updated.

Questions:

The description of the reaction center assignment using Schwaller's method is very brief, and in the current form unclear how it is implemented. Reaction enters can only be assigned to full reactions (you need to have the reactants AND product for that). It is unclear how this can actually work during inference. Given the later corrected results of the 2 papers results I am a bit skeptical how this is actually done, and I would like to see a very clear and long description on what is actually done here.

I also had a look at the code, but could not find the data, or how this additional step is performed in the code.

I am looking forward to discuss this in the rebuttal phase.

______
Edit after discussion:

This reviewer has raised their score to reflect the improvements the authors will make to the manuscript after discussion with the reviewers.
This paper should be published in NeurIPS



**Time Spent Reviewing:**

3

---

> ### Author Response · Authors · 2021-08-10
> **Thank you**
>
> Thank you for the very helpful reviews and your time spent reviewing the paper AND the code. We really appreciate your constructive questions about clarifying the reaction center and generalized dual model.  That is very important for us.
>
> **Re: reaction center, generalized dual model & information leak** \
> Please refer to the comment with title "information leak issue reported in RetroXpert: Updated results". We really appreciate your comments on using inferred reaction centers. We agree with you. We will delete the generalized model section, and only use the dual model as our main results.
>
> For discussion purpose, one way to use a generalized model during testing is to infer the reaction center by the given product and inferred reactants by a backward model trained on data without reaction center. With this inferred reaction center, we can potentially run generalized dual model during testing.

---

### Official Review · Reviewer_EqsE · 2021-07-16

**Rating:** 7
**Confidence:** 5

**Summary:**

This submission tackles the problem of one-step retrosynthesis, a classic problem in chemistry, through the lens of energy-based models. The framing of the problem in this manner is elegant. The quantitative results may need to be clarified or resolved to ensure that the comparison to prior SOTA is fair (i.e., not using an additional model or leaking data of the ground truth reactants).

**Limitations And Societal Impact:**

Limitations are not discussed.

**Main Review:**


The framework presented here is an elegant way to think about the complementarity between retrosynthesis and forward synthesis. It is described well.

1.	This is a revised submission from ICLR 2021 where the authors were unfortunately punished for lacking comparisons to SOTA RetroXpert and GraphRetro, both of which ended up having data leakages that inflated performance by a large margin. Please refer to https://github.com/uta-smile/RetroXpert for an explanation of the leak. Were the sequence-based models provided with atom-mapped SMILES or unmapped SMILES? Were the product and reactant SMILES strings recanonicalized after stripping atom mapping using RDKit? The current SOTA top-1 accuracy when reaction type is unknown is closer to ~53% using the revised RetroXpert and GraphRetro results. The high results reported in Table 2 may be explained by the fact that the “Generalized” model implicitly uses information about the true reactants (see comment below).

2.	The notation in Equation 6 is confusing to me; is $p(X|y) = \exp(- E_\theta(X,y))$ what the authors intended to convey?

3.	The categorization of methods into template-based and semi-template based has an unclear role to me. Is there a need to draw these strong distinctions? Atom mapping can be approximated through maximum common substructure heuristics, so it is “free” in a sense. None of the datasets used for benchmarking these tools relied on manual labeling (cf. line 226). This limitation of methods requiring atom mapping is overstated.

4.	The setting displayed in Table 1 where the inferred reaction center is provided is an unusual choice as it does not reflect a real use case for these models in chemistry. Providing the inferred reaction center (which is inferred given knowledge of the true reactants) to the “Generalized” model makes the comparison to previously reported results invalid, but would explain the 69.9% reported in Table 2 in the absence of the data leak.

If the quantitative results are clarified or resolved to ensure that the comparison to prior SOTA is fair (i.e., not using an additional model or leaking data of the ground truth reactants) I believe this paper is deserving of a 7 (accept).

-----
I have raised my score to 7, as the authors have addressed the (potential) information leak and atypical evaluation with the generalized model

**Time Spent Reviewing:**

2.5

---

> ### Author Response · Authors · 2021-08-10
> **Re: Thank you**
>
> Thank you very much for the insightful suggestions and your time spent reviewing this work. We really appreciate that. Thank you for requesting clarification of the quantitative results compared with prior SOTA methods. With your suggestions, our work can improve sufficiently.
>
> **Re: eq6**: \
> eq6 is to show how we design the energy function for the dual model . The design of energy function is free of choice, but the choice will influence how flexible the model is and how well the model can be trained.
>
> **Re: Categorization of methods**: \
> Yes, we agree that atom mapping can be approximated and relatively easy to be used in practice. The main reason we set this distinction is to compare the learning power of each method with input features of the same level of usefulness (similar with separately reporting performance of models trained with or without reaction type ).
>
> **Re: Reaction center and information leak**: \
> We agree with you and please refer to this comment "Re "information leak issue reported in RetroXpert": Updated results "
>
> Thank you very much.

---

### Author Response · Authors · 2021-08-10
**Re "information leak issue reported in RetroXpert": Updated results**

 We thank all the reviewers for the thorough and insightful comments. Your reviews benefit our paper significantly. We appreciate that the majority of the reviewers found the paper interesting (though with limitations) and could eventually fit in NeurIPS upon addressing some outstanding concerns. Information leak is the major one.

Thank you very much for pointing out the information leak issue about RetroXpert and GraphRetro.   We only realized the issue after the NeurIPS submission. We apologize for this.

We examine the information leak issue on our methods. We found our “generalized dual model”, which takes the reaction center inferred by rxnmapper as additional input, also has the issue of leaking information. The rxnmapper takes unmapped reactions as input and outputs atom-mapped reactions. We should not use reaction data (include reactants) to infer reaction center during testing.  Therefore, we will delete the generalized dual model section (sec 2.3). We sincerely apologize for this. Instead, we will continue to report the un-generalized version, e.g. dual model which doesn’t incorporate reaction center, as our main results.


We will replace table 2 with the table below.


|      	|                      	|       	|       	| Proposal 	|        	|         	|                           	| Re-rank  	|       	|       	|        	|
|------	|----------------------	|-------	|-------	|----------	|--------	|---------	|---------------------------	|----------	|-------	|-------	|--------	|
| Type 	| Proposal Methods     	| Top 1 	| Top 5 	| Top 10   	| Top 50 	| Top 100 	| Rank Model                	| Top 1    	| Top 3 	| Top 5 	| Top 10 	|
| No   	| Ordered on USPTO     	| 44.4  	| 64.9  	| 69.9     	| 77.2   	| 78      	| Dual trained on Aug USPTO 	| 53.6     	| **70.7**  	| **74.6**  	| **77**     	|
|      	| Ordered on Aug USPTO 	| 53.2  	| 54.7  	| 55.6     	| 60.5   	| 60.5    	|   Dual trained on Aug USPTO                         	| **54.5**     	| 60    	| 60.4  	| 60.5   	|
|      	|                      	|       	|       	|          	|        	|         	| SOTA (RextroXpert)        	| 50.4     	| 61.1  	| 62.3  	| 63.4   	|
| Yes  	| Ordered on USPTO     	| 56    	| 76.1  	| 79.7     	| 85.2   	| 86.4    	| Dual trained on Aug USPTO 	| 65.7     	| **81.9**  	| **84.7**  	| **85.9**   	|
|      	| Ordered on Aug USPTO 	| 64.7  	| 66.5  	| 67.3     	| 69.7   	| 75.7    	|    Dual trained on Aug USPTO                        	| **66.2**     	| 75.1  	| 75.6  	| 75.7   	|
|      	|                      	|       	|       	|          	|        	|         	| SOTA (RextroXpert)        	| 62.1     	| 75.8  	| 78.5  	| 80.9   	|

The results show that compare with SOTA RetroXpert, we have 3.6% gain without reaction type and 4.1% percent gain with reaction type.

We also examined the information leak issue by RetroXpert. We found our methods do not suffer from that information leak because (1) our methods use unmapped reaction as input, unlike RetroXpert using mapped reactions (2) our methods’s inputs are random smiles as augmentation to ensure good performance for transformer, not the canonical smiles which leak information by placing the reacted atoms in the first location of a reaction. We have proved the statement on RetroXpert preprocessed unleaked data, and recover similar results without RetroXpert processing.

---

> ### Comment · Reviewer_2KjZ · 2021-08-12
> **Will the authors also replace both replace table 1 and 2?**
>
> This reviewer wants to thank the authors for their kind reply!
>
> This reviewer believes the authors meant that they replace both table 1 + 2, accordingly, is that correct?
>
> After the proposed changes, this reviewer believes the paper is ready for acceptance at NeurIPS and will adjust their score accordingly.

---

> > ### Author Response · Authors · 2021-08-13
> > **Replace table 1 and table 2**
> >
> > We sincerely thank the reviewer for your valuable time spent in reviewing our paper and rebuttal. We really appreciate your constructive suggestions!
> > Yes, absolutely. We will change both Table 1 and Table 2. We will update Table 1 with the above table (e.g. un-generalized dual model in the rebuttal), and update the corresponding entries in Table 2 (e.g. Dual model, remove "Generalized dual model"). We will also update RetroXpert and GraphRETRO with their latest results.

---

> > > ### Comment · Reviewer_2KjZ · 2021-08-13
> > > **thanks, score is adjusted up**
> > >
> > > thanks. this reviewer has adjusted their score up.

---

### Decision · Program_Chairs · 2021-09-27

**Decision:**

Accept (Poster)

**Comment:**

This paper tries to include all the existing retrosynthesis algorithms into the framework of Energy-Based Model (EBM). Based the this, it derives a dual model to measure the energy of a pair (Reactants, Product), which depends on both the backward, forward energy function and their consistency. Experiments demonstrate this will improve the top-1 accuracy by 1.3~1.5%, compared with only using the backward model.